# Gotta Go Fast with Score-Based Generative Models

**Alexia Jolicoeur-Martineau**
Department of Computer Science
University of Montreal

**Ke Li**[†]
Department of Computer Science
Simon Fraser University

**Rémi Piché-Taillefer**[†]
Department of Computer Science
University of Montreal

**Tal Kachman**[†]
Department of Artificial Intelligence
Radboud University & Donders Institute

**Ioannis Mitliagkas**
Department of Computer Science
University of Montreal

## Abstract

Score-based (denoising diffusion) generative models have recently gained a lot of success in generating realistic and diverse data. These approaches define a forward diffusion process for transforming data to noise and generate data by reversing it. Unfortunately, current score-based models generate data very slowly due to the sheer number of score network evaluations required by numerical SDE solvers. In this work, we aim to accelerate this process by devising a more efficient SDE solver. Our solver requires only two score function evaluations per step, rarely rejects samples, and leads to high-quality samples. Our approach generates data 2 to 10 times faster than EM while achieving better or equal sample quality. For high-resolution images, our method leads to significantly higher quality samples than all other methods tested. Our SDE solver has the benefit of requiring no step size tuning.

## 1  Introduction

Score-based generative models [Song and Ermon, 2019, Ho et al., 2020, Jolicoeur-Martineau et al., 2020, Piché-Taillefer, 2021] have been very successful at generating data from various modalities[Song et al., 2020a, Chen et al., 2020, Kong et al., 2020, Niu et al., 2020]. These models generally achieve superior performances in terms of quality and diversity than the historically dominant Generative Adversarial Networks (GANs) [Goodfellow et al., 2014]. Although very powerful, score-based models generate data through an undesirably long iterative process; meanwhile, other state-of-the-art methods such as GANs generate data from a single forward pass of a neural network. Increasing the speed of the generative process is thus an active area of research.

Existing methods for acceleration Chen et al. [2020], San-Roman et al. [2021], Song et al. [2020a,b] often require considerable step size/schedule tuning and do not always work for both Variance Exploding (VE) and Variance Preserving (VP) processes (the two most popular diffusion processes for score-based models). To improve speed and remove the need for step size/schedule tuning, we propose to solve the reverse diffusion process using SDE solvers with adaptive step sizes.

---

† Equal contribution

35th Conference on Neural Information Processing Systems (NeurIPS 2021), Sydney, Australia.

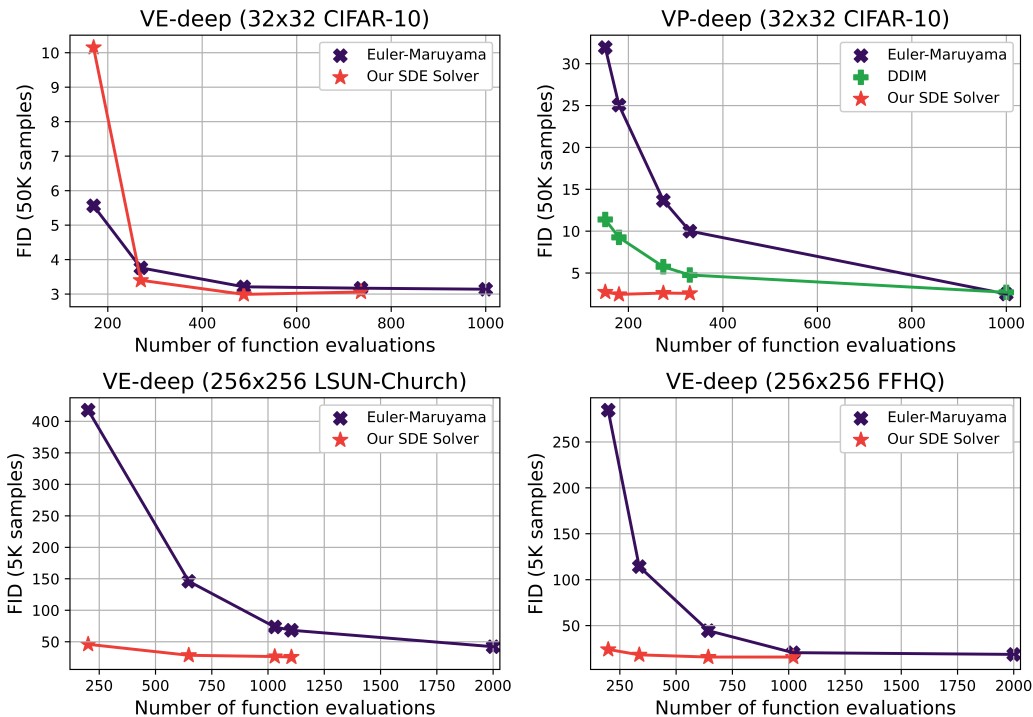

Figure 1: Comparison between our novel SDE solver at various values of error tolerance and Euler-Maruyama for an equal computational budget. We measure speed through the Number of score Function Evaluations (NFE) and the quality of the generated images through the Fréchet Inception Distance (FID; lower is better). See Table 1-2 for more details.

It turns out that off-the-shelf SDE solvers are ill-suited for generative modeling and exhibit either (1) divergence, (2) slower data generation than the baseline, or (3) significantly worse quality than the baseline. This can be attributed to: (1) the extremely high-dimensionality; (2) the high compute of evaluating the score function; (3) the low required precision of the solution because we are satisfied as long as the error is not perceptible (e.g., one RGB increment on an image).

We devise our own SDE solver with these features in mind, resulting in an algorithm that can get around the problems encountered by off-the-shelf solvers.

## 2 Background

### 2.1 Score-based modeling with SDEs

Let $\mathbf{x}(0) \in \mathbb{R}^d$ be a sample from the data distribution $p_{\text{data}}$. The sample is gradually corrupted over time through a Forward Diffusion Process (FDP), a common type of Stochastic Differential Equation (SDE):

$$\mathrm{d}\mathbf{x} = f(\mathbf{x}, t)\mathrm{d}t + g(t)\mathrm{d}\mathbf{w}, \tag{1}$$

where $f(\mathbf{x}, t) : \mathbb{R}^d \times \mathbb{R} \to \mathbb{R}^d$ is the drift, $g(t) : \mathbb{R} \to \mathbb{R}$ is the diffusion coefficient and $\mathbf{w}(t)$ is the Wiener process indexed by $t \in [0, 1]$. Data points and their probability distribution evolve along the trajectories $\{\mathbf{x}(t)\}_{t=0}^1$ and $\{p_t(\mathbf{x})\}_{t=0}^1$ respectively, with $p_0 \equiv p_{\text{data}}$. The functions $f$ and $g$ are chosen such that $\mathbf{x}(1)$ be approximately Gaussian and independent from $\mathbf{x}(0)$. Inference is achieved by reversing this diffusion, drawing $\mathbf{x}(1)$ from its Gaussian distribution and solving the Reverse Diffusion Process (RDP) equal to:

$$\mathrm{d}\mathbf{x} = \left[ f(\mathbf{x}, t) - g(t)^2 \nabla_{\mathbf{x}} \log p_t(\mathbf{x}) \right] \mathrm{d}t + g(t)\mathrm{d}\bar{\mathbf{w}}, \tag{2}$$

where $\nabla_{\mathbf{x}} \log p_t(\mathbf{x})$ is referred to as the score of the distribution at time $t$ [Hyvärinen, 2005] and $\bar{\mathbf{w}}(t)$ is the Wiener process in which time flows backward [Anderson, 1982]. The RDP requires the score

(or $p_t$), which can be estimated by a neural network (referred to as the score network) by optimizing the following objective:

$$\mathcal{L}(\theta) = \mathbb{E}_{\mathbf{x}(t)\sim p(\mathbf{x}(t)|\mathbf{x}(0)),\mathbf{x}(0)\sim p_{\text{data}}} \left[ \frac{\lambda(t)}{2} \left\| s_\theta(\mathbf{x}(t),t) - \nabla_{\mathbf{x}(t)} \log p_t(\mathbf{x}(t)|\mathbf{x}(0)) \right\|_2^2 \right], \quad (3)$$

where $\lambda(t) : \mathbb{R} \to \mathbb{R}$ is chosen to be inversely proportional to $\mathbb{E}\left[ \left\| \nabla_{\mathbf{x}(t)} \log p_t(\mathbf{x}(t)|\mathbf{x}(0)) \right\|_2^2 \right]$.

There are two primary choices for the FDP in the literature, which we discuss below.

## 2.2 Variance Exploding (VE) process

The Variance Exploding (VE) process consists in the following FDP:

$$d\mathbf{x} = \sqrt{\frac{d\left[\sigma^2(t)\right]}{dt}} d\mathbf{w}.$$

Its associated transition kernel is:

$$\mathbf{x}(t)|\mathbf{x}(0) \sim \mathcal{N}(\mathbf{x}(0), [\sigma^2(t) - \sigma^2(0)]\mathbf{I}) \approx \mathcal{N}(\mathbf{x}(0), \sigma^2(t)\mathbf{I}).$$

Given a large enough $\sigma^2(1)$, $\mathbf{x}(1)$ is approximately distributed as $\mathcal{N}(\mathbf{0}, \sigma^2(1)\mathbf{I})$.

## 2.3 Variance Preserving (VP) process

The Variance Preserving (VP) process consists in the following FDP:

$$d\mathbf{x} = -\frac{1}{2}\beta(t)\mathbf{x}dt + \sqrt{\beta(t)}d\mathbf{w}.$$

Its associated transition kernel is:

$$\mathbf{x}(t)|\mathbf{x}(0) \sim \mathcal{N}(\mathbf{x}(0)\, e^{-\frac{1}{2}\int_0^t \beta(s)ds}, (1 - e^{-\int_0^t \beta(s)ds})\, \mathbf{I}).$$

$\mathbf{x}(1)$ is approximately distributed as $\mathcal{N}(\mathbf{0}, \mathbf{I})$ and does not depend on $\mathbf{x}(0)$.

## 2.4 Solving the Reverse Diffusion Process (RDP)

There are many ways to solve the RDP; the most basic one being Euler-Maruyama (EM) [Kloeden and Platen, 1992], the SDE analog to Euler's method for solving ODEs. Song et al. [2020a] also proposed *Reverse-Diffusion*, which consists in ancestral sampling [Ho et al., 2020] with the same discretization used in the FDP. With the Reverse-Diffusion, [Song et al., 2020a] obtained poor results unless applying an additional Langevin dynamics step after each Reversion-Diffusion step; this approach is only heuristically motivated.

# 3 Efficient Method for Solving Reverse Diffusion Processes

## 3.1 Integration method

To dynamically adjust the step size over time, thereby gaining speed over a fixed-step size algorithm, two integration methods are employed. A lower-order ($\mathbf{x}'$) method is used conjointly with a higher-order ($\mathbf{x}''$) one. The local error $E(\mathbf{x}', \mathbf{x}'') = \mathbf{x}' - \mathbf{x}''$ is used to determine how stable the lower-order method is at the current step size; the closer to zero, the more appropriate the step size is. From this information, the step size can be dynamically adjusted and $x'$ can be accept or rejected.

While higher-order solvers may achieve lower discretization errors, they require more function evaluations, and the improved precision might not be worth the increased computation cost [Lehn et al., 2002, Lamba, 2003]. For this reason, we use EM (order 1) and stochastic Improved Euler [Roberts, 2012] (order 2) for the lower and higher integration methods . This approach only requires two score function evaluations; however, it leads to images of poor quality. Thankfully, by using extrapolation (accepting $\mathbf{x}''$ instead of $\mathbf{x}'$ as our proposal), we improve over the baseline approach.

## 3.2 Tolerance

In ODE/SDE solvers, the local error is divided by a *tolerance* term. We calculate the mixed tolerance through the maximum of the current and previous sample:

$$\boldsymbol{\delta}(\mathbf{x}', \mathbf{x}'_{prev}) = \max(\epsilon_{abs}, \epsilon_{rel} \max(|\mathbf{x}'|, |\mathbf{x}'_{prev}|)). \tag{4}$$

For image generation, we can set $\epsilon_{abs}$ a priori. During training, images are represented as floating-point tensors with range $[y_{min}, y_{max}]$, but when evaluated, they are converted to 8-bit color images: scaled to $[0, 255]$ and converted to the nearest integer. This means that an absolute tolerance $\epsilon_{abs} = \frac{y_{max} - y_{min}}{256}$ corresponds to tolerating local errors of at most one color (e.g., $x'_{ij}$ with Red=5 and $x''_{ij}$ with Red=6 is accepted, while Red=7 is not) channel-wise. One-color differences are not perceptible and should not influence the metrics used for evaluating the generated images. To control speed/quality, we vary $\epsilon_{rel}$.

## 3.3 Norm of the scaled error

The scaled error (the error scaled by the mixed tolerance) is calculated as

$$E_q = \left\| \frac{\mathbf{x}' - \mathbf{x}''}{\boldsymbol{\delta}(\mathbf{x}', \mathbf{x}'_{prev})} \right\|_q.$$

Many algorithms use $q = \infty$ [Lamba, 2003, Rackauckas and Nie, 2017a], where $||\mathbf{x}||_\infty = \max(\mathbf{x}_1, ..., \mathbf{x}_k)$ over all $k$ elements of $\mathbf{x}$. This is highly problematic for high-dimensional SDEs such as those in image-space because a single channel of a single pixel (out of 65536 pixels for a $256 \times 256$ color image) with a large local error will cause the step size to be reduced for all pixels. To that effect, we instead use a scaled $\ell_2$ norm (the $\ell_2$ norm multiplied by $\sqrt{k}$).

## 3.4 Hyperparameters of the dynamic step size algorithm

Upon calculating the scaled error, we accept the proposal $\mathbf{x}''$ if $E_q \leq 1$ and increment the time by $h$. Whether or not it is accepted, we update the next step size $h$ in the usual way:

$$h \leftarrow \min(h_{\max}, \theta h E_q^{-r}),$$

where $h_{\max}$ is the maximum step size, $\theta$ is the safety parameter, and $r$ is an exponent-scaling term. Although ODE theory tells which $r$ is optimal, there is no such theory for SDEs [Rackauckas and Nie, 2017a]. Following Rackauckas and Nie [2017a], we empirically testing values and found $r \in [0.8, 0.9]$ to work well (see Appendix E). We arbitrarily chose $r = 0.9$, $\theta = 0.9$, and $h_{\max}$ as the largest step size possible, namely the remaining time $t$. We have now defined every aspect of the algorithm needed to numerically solve the Equation (2) for images. The resulting algorithm is described in Algorithm 1.

# 4 Experiments

We apply our method to pre-trained VP or VE models from Song et al. [2020a] on CIFAR-10 [Krizhevsky et al., 2009] (32x32) and higher-res (256x256) LSUN-Church [Yu et al., 2015], and Flickr-Faces-HQ (FFHQ) [Karras et al., 2019]. We measure performance through the Fréchet Inception Distance (FID) [Heusel et al., 2017] and the Inception Score (IS) [Salimans et al., 2016], whereas low FID and high IS correspond to higher quality/diversity. We used $\leq 4$ V100 GPUs to run the experiments. Results are presented in Figure 1 (and Tables 1 and 2). Probability Flow [Song et al., 2020a] solves an ODE instead of an SDE using Runge-Kutta 45 [Dormand and Prince, 1980] and denoising diffusion implicit models (DDIM) [Song et al., 2020b] is only defined for VP models.

# 5 Conclusion

We built an SDE solver to generate images of comparable (or better) quality to Euler-Maruyama at a much faster speed. Our approach makes image generation with score-based models more accessible by shrinking the required computational budgets by a factor of 2 to 5×, and presenting a sensible way of compromising quality for additional speed. Nevertheless, data generation remains slow (a few minutes) compared to other generative models, which can generate data in a single forward pass of a neural network.

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

# Appendices

## A  More details on the forward processes

### A.1  Variance Exploding (VE) process

In practice, we let $\sigma(t) = \sigma_{min} \left( \frac{\sigma_{max}}{\sigma_{min}} \right)^t$, where $\sigma_{min} = 0.01$ and $\sigma_{max} \approx \max_i \sum_{j=1}^{N} ||\mathbf{x}^{(i)} - \mathbf{x}^{(j)}||$ is the maximum Euclidean distance between two samples from the dataset $\{\mathbf{x}^{(i)}\}_{i=1}^{N}$ [Song and Ermon, 2020]. Using the maximum Euclidean distance ensures that $\mathbf{x}(1)$ does not depend on $\mathbf{x}(0)$; thus, $\mathbf{x}(1)$ is approximately distributed as $\mathcal{N}(\mathbf{0}, \sigma^2(1)\mathbf{I})$.

### A.2  Variance Preserving (VP) process

In practice, we let $\beta(t) = \beta_{min} + t \left( \beta_{max} - \beta_{min} \right)$, where $\beta_{min} = 0.1$ and $\beta_{max} = 20$. Thus, $\mathbf{x}(1)$ is approximately distributed as $\mathcal{N}(\mathbf{0}, \mathbf{I})$ and does not depend on $\mathbf{x}(0)$.

## B  Algorithm

---

**Algorithm 1** Dynamic step size extrapolation for solving Reverse Diffusion Processes

---

**Require:** $s_\theta, \epsilon_{rel}, \epsilon_{abs}, h_{init} = 0.01, r = 0.9, \theta = 0.9$     $\triangleright$ for images: $\epsilon_{abs} = \frac{y_{max} - y_{min}}{256}$
  $t \leftarrow 1$
  $h \leftarrow h_{init}$
  Initialize $\mathbf{x}$
  $\mathbf{x}'_{prev} \leftarrow \mathbf{x}$
  **while** $t > 0$ **do**
   Draw $\mathbf{z} \sim \mathcal{N}(\mathbf{0}, \mathbf{I})$
   $\mathbf{x}' \leftarrow \mathbf{x} - hf(\mathbf{x}, t) + hg(t)^2 s_\theta(\mathbf{x}, t) + \sqrt{h}g(t)\mathbf{z}$     $\triangleright$ Euler-Maruyama
   $\tilde{\mathbf{x}} \leftarrow \mathbf{x} - hf(\mathbf{x}', t - h) + hg(t - h)^2 s_\theta(\mathbf{x}', t - h) + \sqrt{h}g(t - h)\mathbf{z}$
   $\mathbf{x}'' \leftarrow \frac{1}{2}(\mathbf{x}' + \tilde{\mathbf{x}})$     $\triangleright$ Improved Euler (SDE version)
   $\boldsymbol{\delta} \leftarrow \max(\epsilon_{abs}, \epsilon_{rel} \max(|\mathbf{x}'|, |\mathbf{x}'_{prev}|))$     $\triangleright$ Element-wise operations
   $E_2 \leftarrow \frac{1}{\sqrt{n}} \left\| (\mathbf{x}' - \mathbf{x}'') / \boldsymbol{\delta} \right\|_2$
   **if** $E_2 \leq 1$ **then**     $\triangleright$ Accept
    $\mathbf{x} \leftarrow \mathbf{x}''$     $\triangleright$ Extrapolation
    $t \leftarrow t - h$
    $\mathbf{x}'_{prev} \leftarrow \mathbf{x}'$
   $h \leftarrow \min(t, \theta h E_2^{-r})$     $\triangleright$ Dynamic step size update
  **return** $\mathbf{x}$

---

## C  Tables of the results

Table 1: Number of score Function Evaluations (NFE) / Fréchet Inception Distance (FID) on CIFAR-10 (32x32) from 50K samples

| Method | VP | VP-deep | VE | VE-deep |
|---|---|---|---|---|
| Reverse-Diffusion & Langevin | 1999 / 4.27 | 1999 / 4.69 | 1999 / **2.40** | 1999 / **2.21** |
| Euler-Maruyama | 1000 / **2.55** | 1000 / **2.49** | 1000 / 2.98 | 1000 / 3.14 |
| DDIM | 1000 / 2.86 | 1000 / 2.69 | – | – |
| Ours ($\epsilon_{rel} = 0.01$) | 329 / 2.70 | 330 / 2.56 | 738 / 2.91 | 736 / 3.06 |
| Euler-Maruyama (same NFE) | 329 / 10.28 | 330 / 10.00 | 738 / 2.99 | 736 / 3.17 |
| DDIM (same NFE) | 329 / 4.81 | 330 / 4.76 | – | – |
| Ours ($\epsilon_{rel} = 0.02$) | 274 / 2.74 | 274 / 2.60 | 490 / **2.87** | 488 / **2.99** |
| Euler-Maruyama (same NFE) | 274 / 14.18 | 274 / 13.67 | 490 / 3.05 | 488 / 3.21 |
| DDIM (same NFE) | 274 / 5.75 | 274 / 5.74 | – | – |
| Ours ($\epsilon_{rel} = 0.05$) | 179 / **2.59** | 180 / **2.44** | 271 / 3.23 | 270 / 3.40 |
| Euler-Maruyama (same NFE) | 179 / 25.49 | 180 / 25.05 | 271 / 3.48 | 270 / 3.76 |
| DDIM (same NFE) | 179 / 9.20 | 180 / 9.25 | – | – |
| Ours ($\epsilon_{rel} = 0.10$) | 147 / 2.95 | 151 / 2.73 | 170 / 8.85 | 170 / 10.15 |
| Euler-Maruyama (same NFE) | 147 / 31.38 | 151 / 31.93 | 170 / 5.12 | 170 / 5.56 |
| DDIM (same NFE) | 147 / 11.53 | 151 / 11.38 | – | – |
| Ours ($\epsilon_{rel} = 0.50$) | 49 / 72.29 | 48 / 82.42 | 52 / 266.75 | 50 / 307.32 |
| Euler-Maruyama (same NFE) | 49 / 92.99 | 48 / 95.77 | 52 / 169.32 | 50 / 271.27 |
| DDIM (same NFE) | 49 / 37.24 | 48 / 38.71 | – | – |
| Probability Flow (ODE) | 142 / 3.11 | 145 / 2.86 | 183 / 7.64 | 181 / 5.53 |

Table 2: Number of score Function Evaluations (NFE) / Fréchet Inception Distance (FID) on LSUN-Church (256x256) and FFHQ (256x256) from 5K samples

| Method | VE (Church) | VE (FFHQ) |
|---|---|---|
| Reverse-Diffusion & Langevin | 3999 / 29.14 | 3999 / 16.42 |
| Euler-Maruyama | 2000 / 42.11 | 2000 / 18.57 |
| Ours ($\epsilon_{rel} = 0.01$) | 1104 / **25.67** | 1020 / **15.68** |
| Euler-Maruyama (same NFE) | 1104 / 68.24 | 1020 / 20.45 |
| Ours ($\epsilon_{rel} = 0.02$) | 1030 / **26.46** | 643 / **15.67** |
| Euler-Maruyama (same NFE) | 1030 / 73.47 | 643 / 44.42 |
| Ours ($\epsilon_{rel} = 0.05$) | 648 / 28.47 | 336 / 18.07 |
| Euler-Maruyama (same NFE) | 648 / 145.96 | 336 / 114.23 |
| Ours ($\epsilon_{rel} = 0.10$) | 201 / 45.92 | 198 / 24.02 |
| Euler-Maruyama (same NFE) | 201 / 417.77 | 198 / 284.61 |
| Probability Flow (ODE) | 434 / 214.47 | 369 / 135.50 |

## D  DifferentialEquations.jl

Here, we report the preliminary experiments we ran with the *DifferentialEquations.jl* Julia package [Rackauckas and Nie, 2017b] before devising our own SDE solver. As can be seen, most methods either did not converge (with warnings of "instability detected") or converged, but were much slower than Euler-Maruyama. The only promising method was Lamba's method [Lamba, 2003]. Note that an algorithm has strong-order $p$ when the local error from $t$ to $t + h$ is $\mathcal{O}(h^{p+1})$.

Table 3: Short experiments with various SDE solvers from *DifferentialEquations.jl* on the VP model with a small mini-batch.

| Method | Strong-Order | Adaptive | Speed |
|---|---|---|---|
| Euler-Maruyama (EM) | 0.5 | No | Baseline speed |
| SOSRA [Rößler, 2010] | 1.5 | Yes | 5.92 times **slower** |
| SRA3 [Rößler, 2010] | 1.5 | Yes | 6.93 times **slower** |
| Lamba EM (default) [Lamba, 2003] | 0.5 | Yes | Did not converge |
| Lamba EM (atol=1e-3) [Lamba, 2003] | 0.5 | Yes | 2 times **faster** |
| Lamba EM (atol=1e-3, rtol=1e-3) [Lamba, 2003] | 0.5 | Yes | 1.27 times **faster** |
| Euler-Heun | 0.5 | No | 1.86 times **slower** |
| Lamba Euler-Heun [Lamba, 2003] | 0.5 | Yes | 1.75 times **faster** |
| SOSRI [Rößler, 2010] | 1.5 | Yes | 8.57 times **slower** |
| RKMil (at various tolerances) [Kloeden and Platen, 1992] | 1.0 | Yes | Did not converge |
| ImplicitRKMil [Kloeden and Platen, 1992] | 1.0 | Yes | Did not converge |
| ISSEM | 0.5 | Yes | Did not converge |

# E   Effects of modifying Algorithm 1

Table 4: Effect of different settings on the [Inception score (IS) / Fréchet Inception Distance (FID) / Number of score Function Evaluations (NFE)] from 10k samples (with mini-batches of 1k samples) with the VP - CIFAR10 model.

| Change(s) in Algorithm 1 | IS | FID | NFE |
|---|---|---|---|
| No change $\left[q = 2, r = 0.9, \delta(\mathbf{x}', \mathbf{x}'_{prev})\right]$ | 9.38 | 4.70 | 3972 |
| *Small modifications* | | | |
| $\delta(\mathbf{x}')$ | 9.26 | 4.69 | 4166 |
| No Extrapolation (thus, using Euler–Maruyama) | 9.58 | 11.73 | 3978 |
| $q = \infty$ | 9.48 | 4.90 | 14462 |
| $r = .5$ | 9.41 | 4.69 | 4104 |
| $r = .8$ | 9.36 | 4.68 | 3938 |
| $r = 1$ | 9.41 | 4.69 | 4048 |
| *Variations of Lamba [2003] Algorithm* | | | |
| $r = 0.5$, Lamba integration | 7.80 | 52.98 | 1468 |
| $r = 0.5$, Lamba integration, Extrapolation | 7.32 | 64.65 | 1438 |
| $r = 0.5$, Lamba integration, $q = \infty$ | 9.28 | 21.09 | 2360 |
| $r = 0.5$, Lamba integration, $q = \infty$, $\theta = 0.8$ | 9.21 | 18.82 | 2346 |

As can be seen, most chosen settings lead to better results. However, $r$ seems to have little impact on the FID. Still, using $r \in [0.8, 0.9]$ lead to a little bit less score function evaluations and sometimes lead to lower FID.

Table 5: Effect of different settings on the [Inception score (IS) / Fréchet Inception Distance (FID) / Number of score Function Evaluations (NFE)] from 10k samples (with mini-batches of 1k samples) with the VE - CIFAR10 model.

| Change(s) in Algorithm 1 | IS | FID | NFE |
|---|---|---|---|
| No change $\left[q = 2, r = 0.9, \delta(\mathbf{x}', \mathbf{x}'_{prev})\right]$ | 9.39 | 4.89 | 8856 |
| Small modifications | | | |
| $\delta(\mathbf{x}')$ | 9.39 | 4.99 | 17514 |
| No Extrapolation (thus, using Euler–Maruyama) | 9.58 | 6.57 | 8802 |
| $q = \infty$ | 9.41 | 5.03 | 39500 |
| $r = 0.5$ | 9.47 | 4.87 | 9594 |
| $r = 0.8$ | 9.45 | 4.84 | 8952 |
| $r = 1$ | 9.43 | 4.93 | 8784 |
| Variations of Lamba [2003] Algorithm | | | |
| $r = 0.5$, Lamba integration | 9.08 | 18.28 | 2492 |
| $r = 0.5$, Lamba integration, Extrapolation | 3.70 | 169.78 | 2252 |
| $r = 0.5$, Lamba integration, $q = \infty$ | 9.42 | 6.80 | 5886 |
| $r = 0.5$, Lamba integration, $q = \infty$, $\theta = 0.8$ | 9.35 | 6.20 | 2970 |

## F Implementation Details

We started from the original code by Song et al. [2020a] but changed a few settings concerning the SDE solving. This creates some very minor difference between their reported results and ours. For the VP and VP-deep models, we obtained 2.55 and 2.49 instead of the original 2.55 and 2.41 for the baseline method (EM). For the VE and VE-deep models, we obtained 2.40 and 2.21 instead of the original 2.38 and 2.20 for the baseline method (Reverse-Diffusion with Langevin).

When solving the SDE, time followed the sequence $t_0 = 1$, $t_i = t_{i-1} - \frac{1-\epsilon}{N}$, where $N = 1000$ for CIFAR-10, $N = 2000$ for LSUN, $\epsilon = 1e - 3$ for VP models, and $\epsilon = 1e - 5$ for VE models.

Meanwhile, the actual step size $h$ used in the code for Euler-Maruyama (EM) was equal to $\frac{1}{N}$. Thus, there was a negligible difference between the step size used in the algorithm ($h = \frac{1}{N}$) and the actual step size implied by $t$ ($h = \frac{1-\epsilon}{N}$). Note that this has little to no impact.

The bigger issue is at the last predictor step was going from $t = \epsilon$ to $t = \epsilon - \frac{1}{N} < 0$. Thus, $t$ was made negative. Furthermore the sample was denoised at $t < 0$ while assuming $t = \epsilon$. There are two ways to fix this issue: 1) take only a step from $t = \epsilon$ to $t = 0$ and do not denoise (since you cannot denoise with the incorrect $t$ or with $t = 0$), or 2) stop at $t = \epsilon$ and then denoise. Since denoising is very helpful, we took approach 2; however, both approaches are sensible.

Finally, denoising was not implemented correctly before. Denoising was implemented as one predictor step (Reverse-Diffusion or EM) without adding noise. This corresponds to:

$$\mathbf{x} \leftarrow \mathbf{x} - h \left[ f(\mathbf{x}, t) - g(t)^2 \nabla_{\mathbf{x}} \log p_t(\mathbf{x}) \right].$$

At the last iteration, this incorrect denoising would be:

$$\mathbf{x} \leftarrow \mathbf{x} + \frac{d[\sigma^2(t)]}{dt} \frac{1}{N} \nabla_{\mathbf{x}} \log p_t(\mathbf{x})$$

$$= \mathbf{x} + \frac{\sigma_{min}}{N} \sqrt{2 \log \left( \frac{\sigma_{max}}{\sigma_{min}} \right)} \nabla_{\mathbf{x}} \log p_t(\mathbf{x})$$

$$\approx \mathbf{x}$$

for VE and

$$\mathbf{x} \leftarrow \mathbf{x} + \frac{\sqrt{\beta_{min}}}{N} \nabla_{\mathbf{x}} \log p_t(\mathbf{x})$$

$$\approx \mathbf{x}$$

for VP.

Meanwhile, the correct way to denoise based on Tweedie formula [Efron, 2011] is:

$$\mathbf{x} \leftarrow \mathbf{x} + \text{Var}[\mathbf{x}(t)|\mathbf{x}(0)] \nabla_{\mathbf{x}} \log p_t(\mathbf{x}),$$

where $\text{Var}[\mathbf{x}(t)|\mathbf{x}(0)]$ is the variance of the transition kernel: $\text{Var}[\mathbf{x}(t)|\mathbf{x}(0)] = \sigma_{min} = 0.01$ for VE and $\text{Var}[\mathbf{x}(t)|\mathbf{x}(0)] = 1$. This means that the correct Tweedie formula corresponds to

$$\mathbf{x} \leftarrow \mathbf{x} + 0.01^2 \nabla_{\mathbf{x}} \log p_t(\mathbf{x})$$

$$\approx \mathbf{x}$$

for VE and

$$\mathbf{x} \leftarrow \mathbf{x} + \nabla_{\mathbf{x}} \log p_t(\mathbf{x})$$

for VP.

As can be seen, denoising has a very small impact on VE so the difference between the correct and incorrect denoising is minor. Meanwhile, for VP the incorrect denoising lead to a tiny change, while the correct denoising lead to a large change. In practice, we observe that changing the denoising method to the correct one does not significantly affect the FID with VE, but lowers down the FID significantly with VP.

## G    Inception Score on CIFAR-10

Table 6: Inception Score on CIFAR-10 (32x32) from 50K samples

| Method | VP | VP-deep | VE | VE-deep |
|---|---|---|---|---|
| Reverse-Diffusion & Langevin | 9.94 | 9.85 | 9.86 | 9.83 |
| Euler-Maruyama | 9.71 | 9.73 | 9.49 | 9.31 |
| Ours ($\epsilon_{rel} = 0.01$) | 9.46 | 9.54 | 9.50 | 9.48 |
| Ours ($\epsilon_{rel} = 0.02$) | 9.51 | 9.48 | 9.57 | 9.50 |
| Ours ($\epsilon_{rel} = 0.05$) | 9.50 | 9.61 | 9.64 | 9.63 |
| Ours ($\epsilon_{rel} = 0.10$) | 9.69 | 9.64 | 9.87 | 9.75 |
| Probability Flow (ODE) | 9.37 | 9.33 | 9.17 | 9.32 |

## H    Stability and Bias of the Numerical Scheme

The following constructions rely on the underlying assumption of the stochastic dynamics being driven by a wiener process. More so, we also assume that the Brownian motion is time symmetrical. Both assumptions are consistent and widely used in the literature; for example, see [Gardiner, 2009] [Arnold, 1974].

The method described in Algorithm 1 gives us a significant speedup in terms of computing time and actions. Albeit the speed up comes from a piece-wise step in the algorithm combining the traditional Euler Maruyama (EM) with a form of adaptive step size predictor-corrector. Here we show that both the stability and the convergence of the EM scheme are conserved by introducing the extra adaptive stepsize of our new scheme. As a first step, we define the stability and bias in a Stochastic Differential Equation (SDE) numerical solution.

We denote $\Re(\lambda)$ as the real value of a complex-valued $\lambda$.

The linear test SDE is defined in the following way:

$$\mathrm{d}\mathbf{x}_t = \lambda \mathbf{x}_t \mathrm{d}t + \sigma \mathrm{d}\mathbf{w}_t \tag{5}$$

with its numerical counterpart

$$\mathbf{y}_{n+1} = \Re(h\lambda)\mathbf{y}_n + \mathbf{z}_n,$$

where the $\mathbf{z}_n$ are random variables that do not depend on $\mathbf{y}_0, \mathbf{y}_1......\mathbf{y}_n$ or $\lambda$ and the EM scheme is

$$\mathbf{y}_{n+1} = (1 + h\lambda)\mathbf{y}_n + \mathbf{z}_n.$$

A numerical scheme is asymptotically unbiased with step size $h > 0$ if, for a given linear SDE (5) driven by a two-sided Wiener process, the distribution of the numerical solution $\mathbf{y}_n$ converges as $n \to \infty$ to the normal distribution with zero mean and variance $\frac{\sigma^2}{2|\lambda|}$ [Artemiev and Averina, 2011]. This stems from the fact that a solution of a linear SDE (5) is a Gaussian process whenever the initial condition is Gaussian (or deterministic); thus, there are only two moments that control the bias in the algorithm:

$$\lim_{n\to\infty} \mathbb{E}[\mathbf{y}_n] = 0, \qquad \lim_{n\to\infty} \mathbb{E}[\mathbf{y}_n^2] = -\frac{\sigma^2}{2|\lambda|}.$$

A numerical scheme with step size $h$ is numerically stable in mean if the numerical solution $\mathbf{y}_n^{(h)}$ applied to a linear SDE satisfies

$$\lim_{n\to\infty} \mathbb{E}[\mathbf{y}_n] = 0,$$

and is stable in mean square [Saito and Mitsui, 1996] if we have that

$$\lim_{h\to 0}\left(\lim_{n\to\infty} \mathbb{E}\left[|\mathbf{y}_n|^2\right]\right) = \frac{\sigma^2}{2\Re(\lambda)}.$$

In what follows, we will trace the criteria for bias through our algorithm and show that it remains unbiased. By construction, the first EM step remains unbiased, while for the RDP, we write down the time reverse Wiener process as

$$\tilde{\mathbf{y}}_{n+1} = (1 + \lambda h)\,\tilde{\mathbf{y}}_n + \tilde{\mathbf{z}}_n$$

in the reverse time steps $h$ i.e., $t - nh, t - 2nh$,

$$\begin{aligned}
\mathbb{E}\left[\tilde{\mathbf{y}}_{n+1}\right] &= (1 + \lambda(t - h))\,\mathbb{E}\left[\tilde{\mathbf{y}}_n\right] \\
&= (1 + \lambda(t - h))\,\mathbb{E}\left[(1 + \lambda(t - h))\,\tilde{\mathbf{y}}_{n-1}\right] \\
&\qquad\vdots \\
&= (1 + \lambda(t - h))^{n+1}\,\mathbb{E}\left[\tilde{\mathbf{y}}_0\right] \\
&= (1 + \lambda(t - h))^{n+1}\,\mathbb{E}\left[\mathbf{y}_0\right].
\end{aligned}$$

Thus, if

$$|1 + \lambda(t - h)| < 1,$$

then

$$\lim_{n \to \infty} \mathbb{E}\left[\mathbf{y}_n^{(h)}\right] = 0.$$

In Algorithm 1, we are performing consecutive steps forward and backwards in time so $t = 2h$ such that

$$|1 + \lambda h| < 1.$$

Thus, the scheme is both numerically stable and unbiased with respect to the mean.

Next, we focus on the numerical solution in mean square:

$$\begin{aligned}
\mathbb{E}\left[\left|\tilde{\mathbf{y}}_{n+1}\right|^2\right] &= |1 + \lambda(t - h)|^2\,\mathbb{E}\left[\left|\tilde{\mathbf{y}}_n\right|^2\right] + \sigma^2 h \\
&= |1 + \lambda(t - h)|^2\left\{|1 + \lambda(t - h)|^2\,\mathbb{E}\left[\left|\tilde{\mathbf{y}}_{n-1}\right|^2\right] + \sigma^2 h\right\} + \sigma^2 h \\
&\qquad\vdots \\
&= |1 + \lambda(t - h)|^{2(n+1)}\,\mathbb{E}\left[|\mathbf{y}_0|\right] + \frac{|1 + \lambda(t - h)|^{2(n+1)} - 1}{2\Re\lambda + |\lambda|^2(t - h)}\sigma^2.
\end{aligned}$$

Under the same assumption of consecutive steps, we have that

$$\mathbb{E}\left[\left|\tilde{\mathbf{y}}_{n+1}\right|^2\right] = |1 + \lambda h|^{2(n+1)}\,\mathbb{E}\left[|\mathbf{y}_0|\right] + \frac{|1 + \lambda h|^{2(n+1)} - 1}{2\Re(\lambda) + |\lambda|^2 h}\sigma^2,$$

$$\lim_{n \to \infty} \mathbb{E}\left[\left|\tilde{\mathbf{y}}_{n+1}\right|^2\right] = -\frac{\sigma^2}{2\Re(\lambda) + |\lambda|^2 h},$$

$$\lim_{h \to 0}\left(\lim_{n \to \infty} \mathbb{E}\left[\left|\tilde{\mathbf{y}}_{n+1}\right|^2\right]\right) = -\frac{\sigma^2}{2\Re(\lambda)}.$$

Assuming the imaginary part of $\lambda$ is null, we have that

$$\lim_{h \to 0}\left(\lim_{n \to \infty} \mathbb{E}\left[\left|\tilde{\mathbf{y}}_{n+1}\right|^2\right]\right) = -\frac{\sigma^2}{2\,|\lambda|}.$$

Thus, the numerical scheme is stable and unbiased in the mean square.

Following the two steps for computation of $\mathbf{x}'$ and $\tilde{\mathbf{x}}$, the step size decreases and does not change size; thus, all the above statements hold, and the entire algorithm is stable and unbiased with respect to both the mean and square mean.

# I Samples

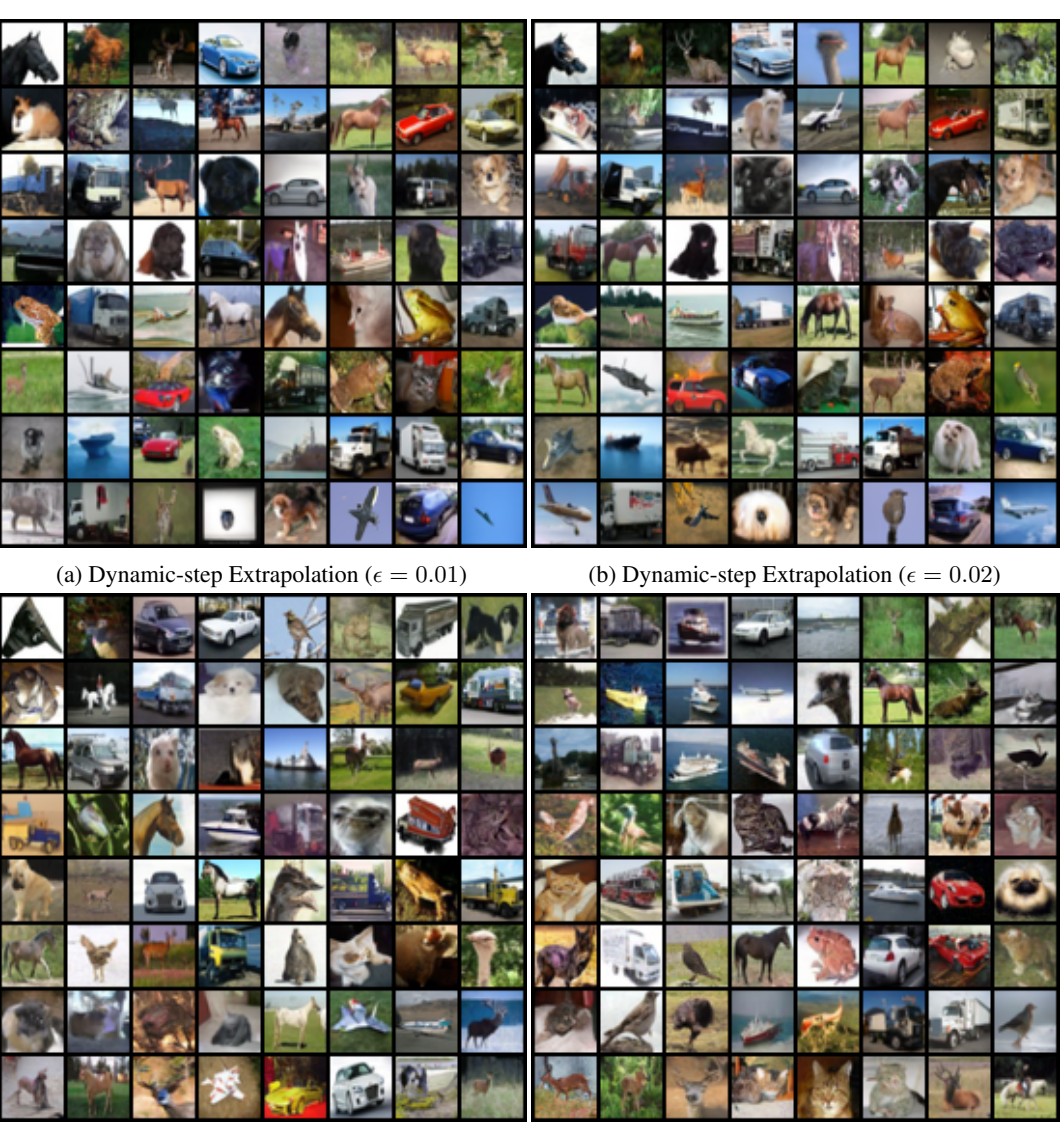

(a) Dynamic-step Extrapolation ($\epsilon = 0.01$)  (b) Dynamic-step Extrapolation ($\epsilon = 0.02$)

(c) Dynamic-step Extrapolation ($\epsilon = 0.05$)  (d) Dynamic-step Extrapolation ($\epsilon = 0.10$)

Figure 2: VP - CIFAR10

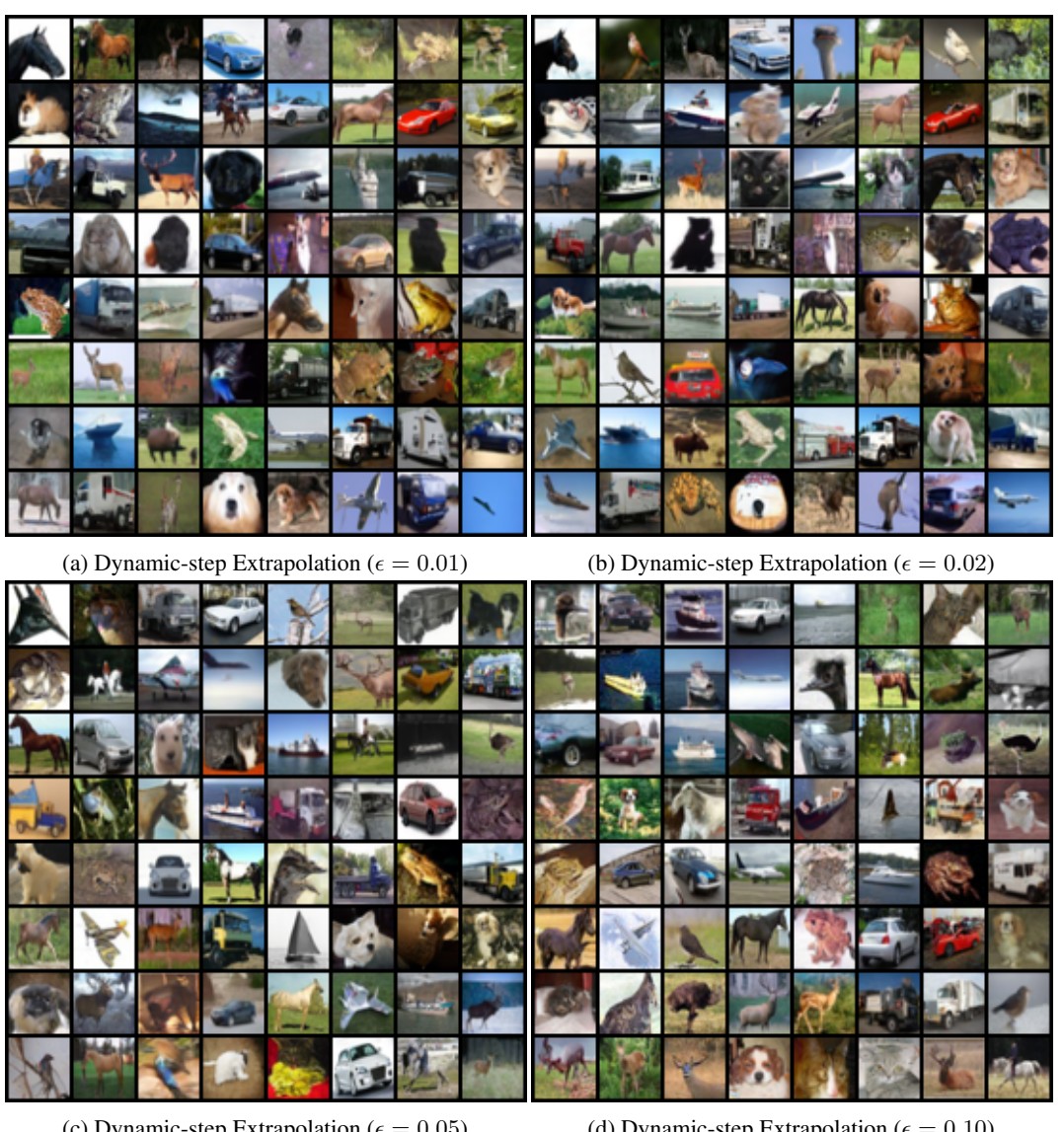

(a) Dynamic-step Extrapolation ($\epsilon = 0.01$)

(b) Dynamic-step Extrapolation ($\epsilon = 0.02$)

(c) Dynamic-step Extrapolation ($\epsilon = 0.05$)

(d) Dynamic-step Extrapolation ($\epsilon = 0.10$)

Figure 3: VP-deep - CIFAR10

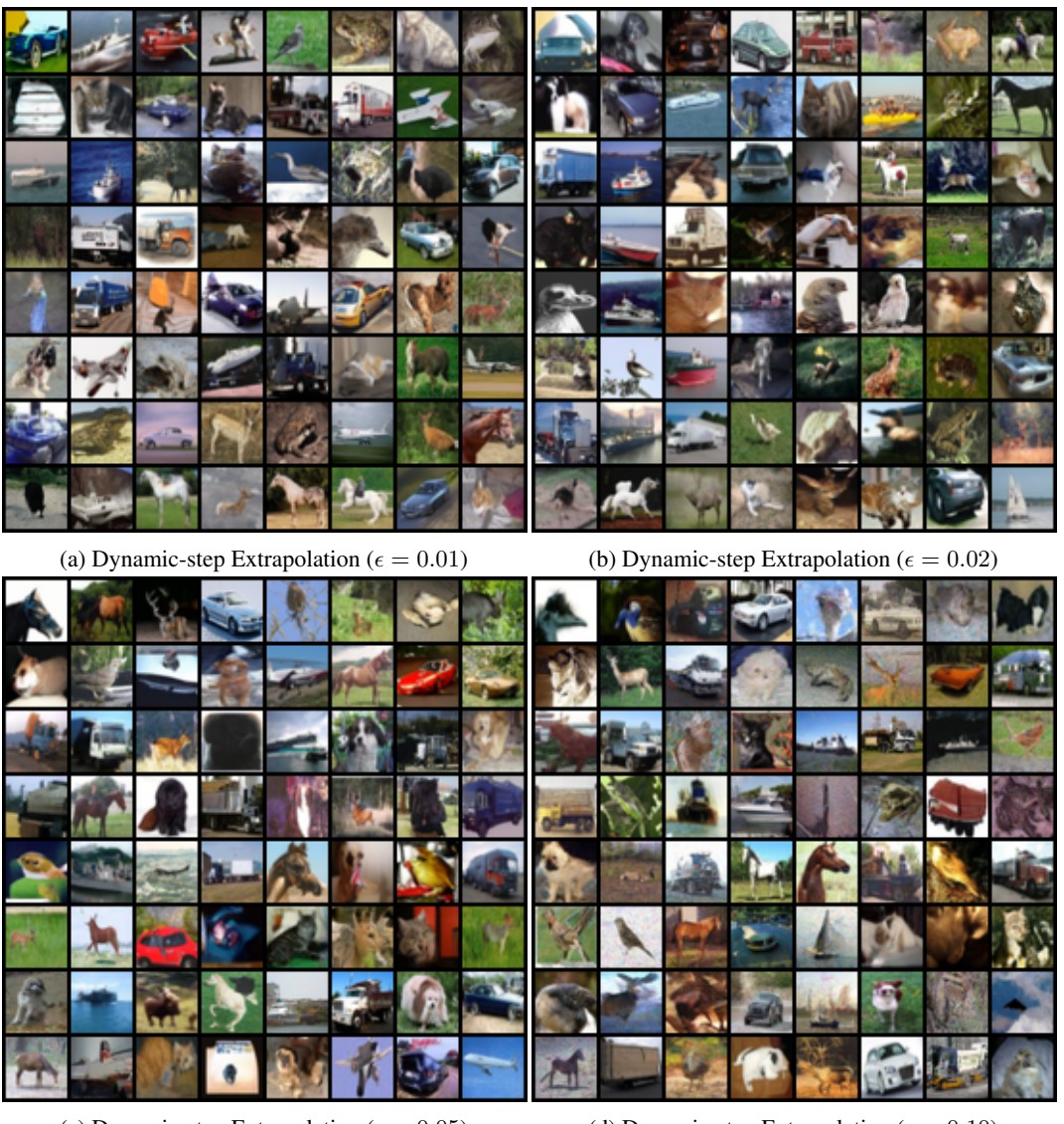

(a) Dynamic-step Extrapolation ($\epsilon = 0.01$)

(b) Dynamic-step Extrapolation ($\epsilon = 0.02$)

(c) Dynamic-step Extrapolation ($\epsilon = 0.05$)

(d) Dynamic-step Extrapolation ($\epsilon = 0.10$)

Figure 4: VE - CIFAR10

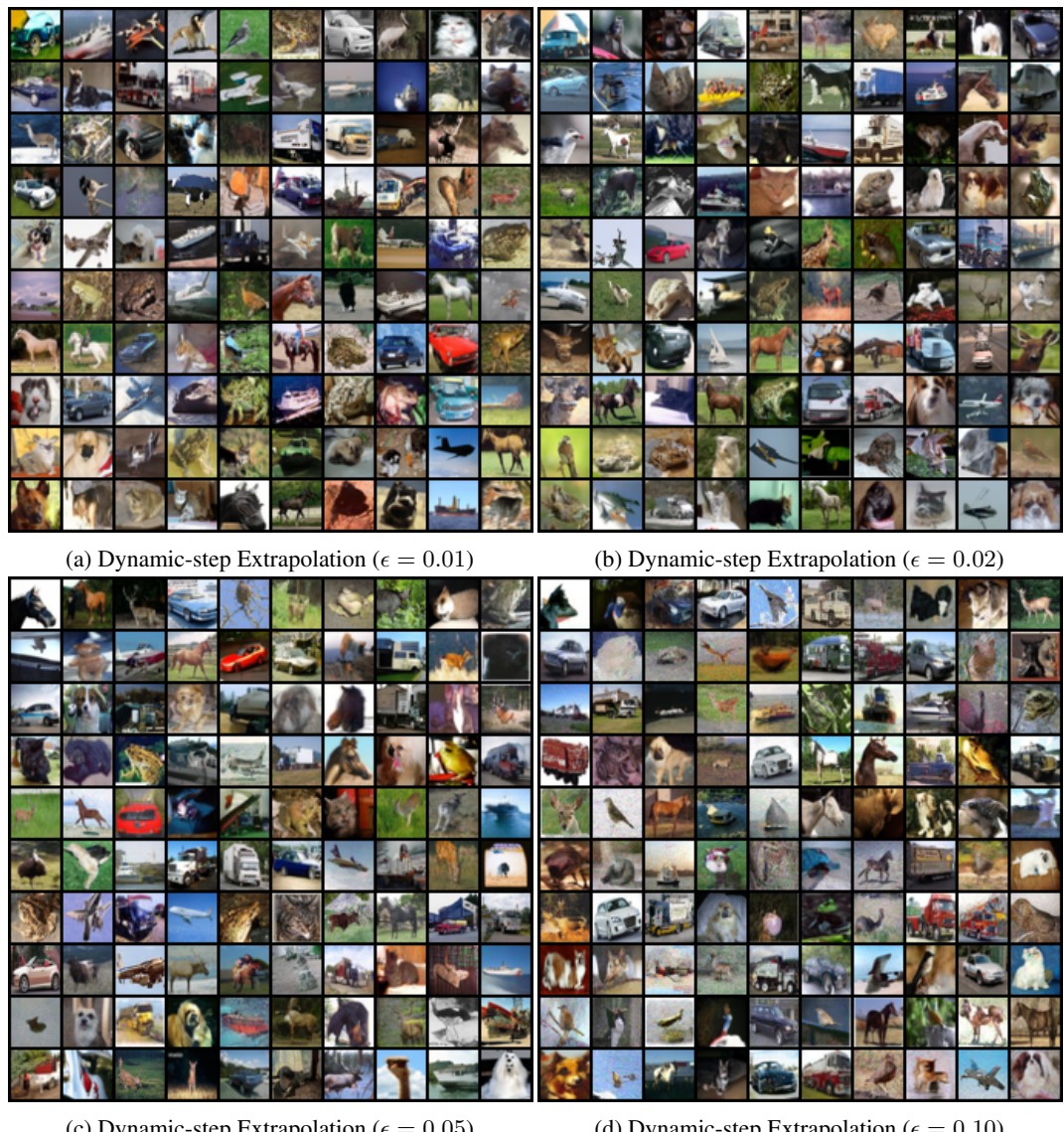

(a) Dynamic-step Extrapolation ($\epsilon = 0.01$)

(b) Dynamic-step Extrapolation ($\epsilon = 0.02$)

(c) Dynamic-step Extrapolation ($\epsilon = 0.05$)

(d) Dynamic-step Extrapolation ($\epsilon = 0.10$)

Figure 5: VE-deep - CIFAR10

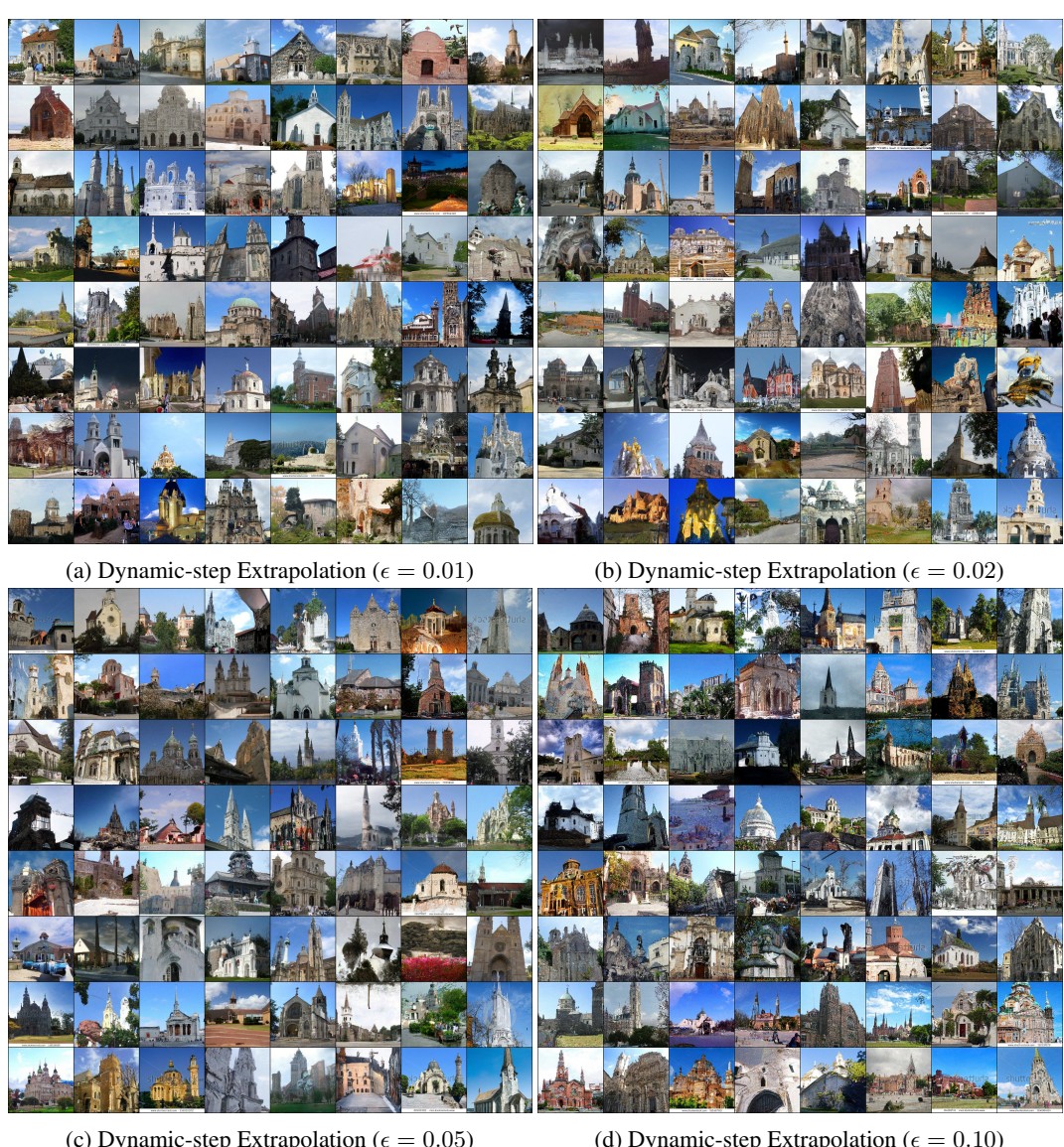

(a) Dynamic-step Extrapolation ($\epsilon = 0.01$)

(b) Dynamic-step Extrapolation ($\epsilon = 0.02$)

(c) Dynamic-step Extrapolation ($\epsilon = 0.05$)

(d) Dynamic-step Extrapolation ($\epsilon = 0.10$)

Figure 6: VE - LSUN-Church (256x256)

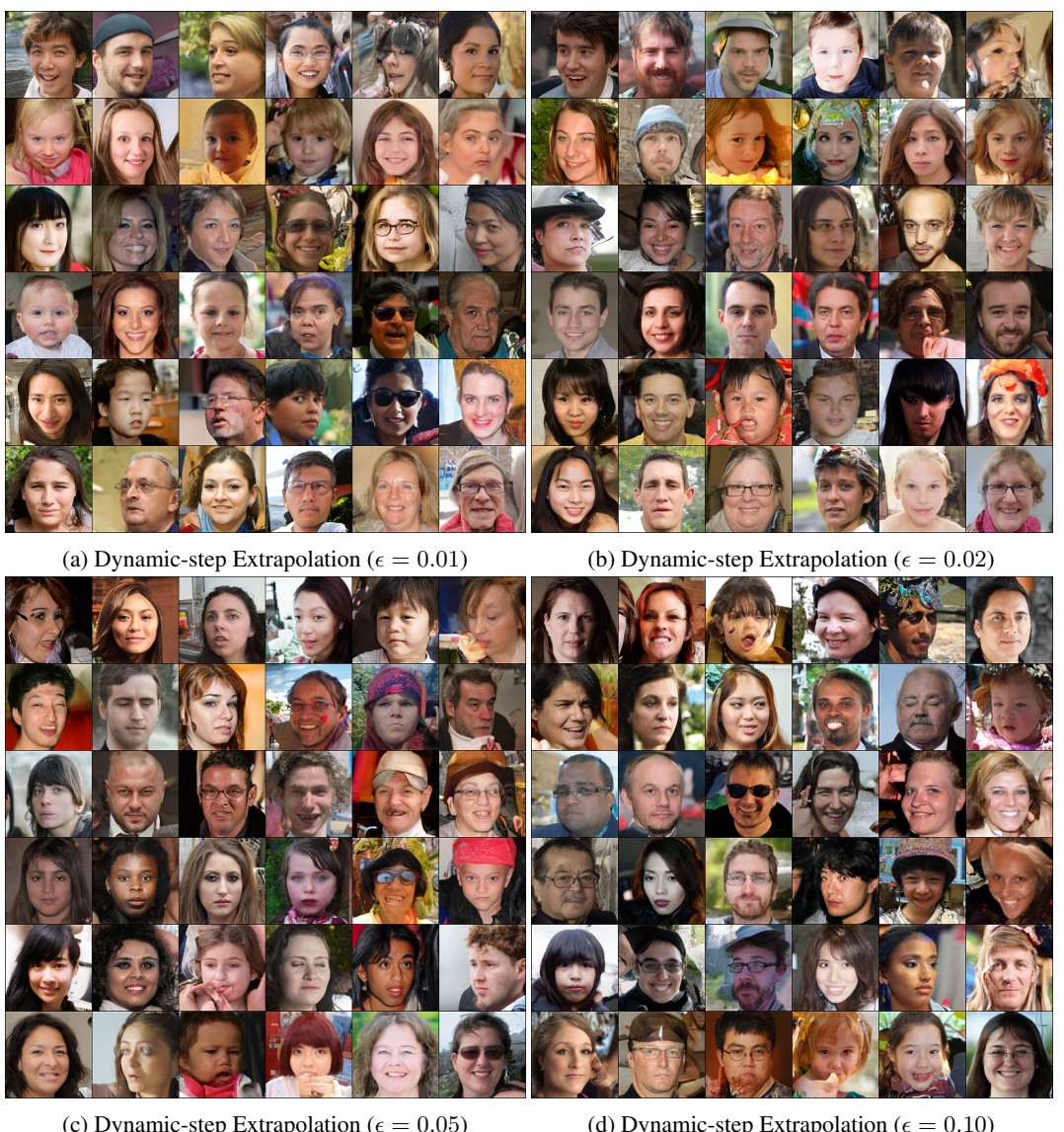

(a) Dynamic-step Extrapolation ($\epsilon = 0.01$)  (b) Dynamic-step Extrapolation ($\epsilon = 0.02$)

(c) Dynamic-step Extrapolation ($\epsilon = 0.05$)  (d) Dynamic-step Extrapolation ($\epsilon = 0.10$)

Figure 7: VE - FFHQ (256x256)

