# OpenReview forum: "Gotta Go Fast with Score-Based Generative Models"
_NeurIPS.cc/2021/Workshop/DLDE — DLDE Workshop -- NeurIPS 2021 Poster_

### Official Review · Reviewer_jG92 · 2021-09-29

**Confidence:** 4

**Review:**

This work investigates the effect of applying different SDE solvers to the RDP of diffusion models. Among various modifications, the authors propose to adjust relative tolerances based on the assumption that their solver will be applied to images, with interesting comments on acceptable errors based on human perception. The authors also propose to combine EM with IEM to obtain a rough estimate of local truncation errors, and use the higher-accuracy step (IEM) to extrapolate.

**Novelty & Significance**

Empirical analyses of numerical methods for solving RDPs of diffusion models are valuable and missing from the current literature. I commend the authors for evaluating a large number of solvers. However, the novelty is quite limited: "our SDE solver" is misleading, as the method really is just a combination of IEM+EM with minor tweaks for images.

The use of l2 norm instead of $\infty$ norm is standard in neural differential equations, see for example solvers in [1] [2]. Seminorms could also be useful in the image domain [3]. This change seems to be the one behind most of the reduction in NFEs for adaptive solvers (Table 4 shows a big difference).

**Questions**:
* Do the results in Table 3 also substitute $\infty$ with L2 norms for adaptive-step algorithms? The comparison is not fair otherwise, as that is not necessarily a component of the numerical method itself. That will also heavily affect "Speed".
* Have the authors evaluated IEM in isolation (as a baseline solver for the RDP), without any of the other modifications? Ablations of this type are necessary to contextualize the results of Figure 1. Perhaps Figure 1 itself could contain baseline (EM) and ablated variants of the proposed solver -- including IEM+EM with EM step, IEM+EM with L2 / $\infty$.
* Have you considered Lamba with L2 norm for step-size control? Lamba seemed promising.

**Minor**:

The title is catchy but completely uninformative. As a completely subjective comment, I'd prefer a clearer title that mentions empirical evaluations of solvers for diffusion models.

line 204: are lead

line 209: This create

[1] torchdiffeq, Chen et al

[2] TorchDyn: A Neural Differential Equations Library, Poli et al

[3] Hey, that’s not an ODE": Faster ODE Adjoints via Seminorms, Kidger et al

**Score:**

3: Good paper

---

### Official Review · Reviewer_9tPU · 2021-10-10

**Confidence:** 4

**Review:**

## Summary

This paper considers score-based generative models and examines one specific step in this method: the step
of sampling the reverse diffusion process. The authors propose to use a modification of the standard
Euler Maruyama SDE solver with an adaptive step size and and compare the generative modeling performance to
the performance achieved when other SDE solvers from the literature are employed.
Notation and terminology are as in Song et al. (2020a).

## Main comments

In my view the work provides an interesting contribution. However, there are several aspects of the work that require
refinement or elaboration before submission to an archival venue. Firstly, since the main goal and contribution
of the work is the speed of the proposed method, the way how this speed is measured requires more details:
- I think it would be good to report in the table also the actual run times for the different methods.
- In the abstract the authos write: "Our solver requires only two score function evaluations". But, according to Algorithm 1, this does not seem correct - the solver requires two evaluations in each step.
- In Figure 1 the authors say that the they measure speed through Number of Function Evalutions, in the supplementary material it appears
to be the number of score function evaluations. It needs to be clarified which of the two is correct.
- At first glance it would appear that the proposed method is slower than Euler-Maruyama, since it does Euler-Maruyama plus additional steps.
To shed light on this, the authors would need to provide - in one of the examples - some (e.g. graphical) insights on the actual choices of
step-sizes chosen by the algorithm and on how often the samples are rejected and redrawn.


## Minor comments

- In Appendix C the authors provide short experiments with other SDE solvers. Some of these methods have the same convergence order, are adaptive and
converge faster than Euler-Maruyama (for instance, Lamba EM, Lamba Euler-Heun). For a thorough comparison the authors would also need to
test one of these methods as a comparison.
- In Appendix E the authors state that the step size for Euler-Maruyama was equal to 1/1000. But to generate Table 1 this must have been
modified, it would be good to be more specific here.
- In Appendix G it seems that the authors do not actually examine the proposed numerical scheme, but rather a backward Euler-Maruyama scheme.
The additional step introduced by the authos does not seem to be analyzed.
- line 27: The sentence here is a bit misleading: the authors should make more clear here that different
method suffer from EITHER of these issues. For instance, Euler-Maruyama achieves the same accuracy.
- The variable n in line 95 needs to be introduced somewhere. n also needs to be specified in Algorithm 1.
- There are a few inaccuracies: a "." is missing in line 4, it should say "Variance Exploding" instead of "Variance Explosing",
in line 39 it should be "diffusion coefficient", in line 77 a ";" needs to be removed, in line 97 a "." is missing, in line 99
there is a superfluous space.
- The choices of sigma and beta in Section 2.2 and 2.3 need to be specified somehwere.
- "DDIM" is not introduced in Table 1.

**Score:**

3: Good paper

---

### Official Review · Reviewer_sChf · 2021-10-11
**Well written with good results**

**Confidence:** 3

**Review:**

### Summary

This work proposes a new SDE Solver specifically adapted to RDP for generative models. The proposed algorithm relies on an integration method that uses both first and second order approximations to compute an adaptive step size, a mixed tolerance term that exploits image structure, a scaled error that stops large individual pixel errors from reducing the step size and a way of tuning the $r$ hyperparameter.

### Review

Overall the paper is well written and organized. The appendix also includes a comprehensive set of experiments and comparisons that thoroughly substantiate the claims presented.  Some potential improvements:
- An wall-time comparison between regular Euler-Maruyama and the variation presented. Although Figure 1 presents a comparison based on function evaluations, it is not obvious whether the algorithmic changes introduced are small  relative to the cost of function evaluation in practice.
- In lines 73-48 the authors discuss the drawbacks of higher order methods in contrast to their approach. But their approach is also a higher order method.
- The authors introduced three independent major changes to Euler-Maruyama in their SDE solver. It would be useful to see a quantitative analysis showing which combination of changes is responsible for the performance improvement observed.

Ultimately, the paper presents a non-trivial improvement for SDE solvers for sample generation.



**Score:**

3: Good paper

---

### Official Review · Reviewer_rjmo · 2021-10-11

**Confidence:** 4

**Review:**

Summary

The paper proposes an SDE solver for the reverse diffusion process in score based generative models.
The paper is well written and within the scope of the workshop.


Questions

Why is the only ablation with respect to EM and not also with IEM? (Fig 1)



**Score:**

3: Good paper

---

### Decision · Program_Chairs · 2021-10-14

**Decision:**

Accept (Poster)

**Comment:**

All reviewers recommend acceptance. The authors might want to consider incorporating suggestions on additional ablations discussed by the reviewers.